# Current Trends in Spent Portable Lithium Battery Recycling

**DOI:** 10.3390/ma16124264

**Published:** 2023-06-08

**Authors:** Zita Takacova, Dusan Orac, Jakub Klimko, Andrea Miskufova

**Affiliations:** Institute of Recycling Technologies, Faculty of Materials, Metallurgy and Recycling, Technical University of Kosice, Letna 9, 04200 Kosice, Slovakia; dusan.orac@tuke.sk (D.O.); jakub.klimko@tuke.sk (J.K.); andrea.miskufova@tuke.sk (A.M.)

**Keywords:** spent portable lithium battery, recycling, pyrometallurgy, hydrometallurgy

## Abstract

This paper provides an overview of the current state of the field in spent portable lithium battery recycling at both the research and industrial scales. The possibilities of spent portable lithium battery processing involving pre-treatment (manual dismantling, discharging, thermal and mechanical-physical pre-treatment), pyrometallurgical processes (smelting, roasting), hydrometallurgical processes (leaching followed by recovery of metals from the leachates) and a combination of the above are described. The main metal-bearing component of interest is the active mass or cathode active material that is released and concentrated by mechanical-physical pre-treatment procedures. The metals of interest contained in the active mass include cobalt, lithium, manganese and nickel. In addition to these metals, aluminum, iron and other non-metallic materials, especially carbon, can also be obtained from the spent portable lithium batteries. The work describes a detailed analysis of the current state of research on spent lithium battery recycling. The paper presents the conditions, procedures, advantages and disadvantages of the techniques being developed. Moreover, a summary of existing industrial plants that are focused on spent lithium battery recycling is included in this paper.

## 1. Introduction

In the European Union (EU), the production of metals from primary raw materials is confronted with a number of problems, including the depletion or complete absence of primary raw material resources and the associated raw material dependency, high extraction costs, low metal content of primary raw materials, etc. As a consequence, EU raw materials policy has started to focus on promoting the supply of own-sourced raw materials and increasing resource efficiency and recycling rates [1]. The main objective is to reduce the dependence of EU countries on imports of raw materials. In 2010, the EU established a list of critical raw materials in terms of their supply risk and economic importance, which is reviewed every three years [1,2]. The current list of critical raw materials from the year 2020 includes 30 critical raw materials instead of the original 14 from the year 2010 [3]. Among them are also the components present in lithium batteries, such as cobalt, lithium and graphite. Material recycling of spent lithium cells is essential in the effort:to obtain a valuable secondary source of the present materials, mainly metals, whose content in spent lithium cells often exceeds their content in primary raw materials;to save natural resources and energy;obtain saleable products with high added value;to achieve self-sufficiency in raw materials, etc.

Material recycling is also important in the development of a circular economy. Implementing the circular economy principle will close the life cycle of spent lithium cells, which consists of several phases, including production, consumption and recycling of spent lithium batteries.

The amount of spent lithium batteries (LiBs) is growing every year as a result of the boom in IT and telecommunication technologies and the expansion of electromobility supported by EU. The current increase in the consumption of lithium cells will be reflected in a rapid increase in the amount of spent lithium cells intended for recycling.

By 2030, the amount of spent LiBs in the European waste market is expected to increase 5–6 times compared with that in 2022 [1]. Currently, mainly portable and industrial LiBs are available on the waste market, but LiBs from electric vehicles (EVs) and hybrid electric vehicles (HEVs) are expected to start increasing rapidly in the near future. However, these batteries have further uses as energy storage sources, which will delay the need for their material recycling for some time. In the EU, a new regulation is in the process of being approved which will replace the existing Directive 2006/66/EC. The new regulation will, for example, increase the minimum limits for the collection and recycling efficiency of spent electrochemical cells and introduce limits for the material recovery of specific metals contained in them, such as cobalt and lithium.

Currently in the EU, the recycling rate of metals present in spent portable LiBs is 22% for cobalt and less than 1% for Li [3], indicating a large potential for increasing. The price of these metals and their content in portable LiBs also play in favor of recycling. While the cobalt content in primary raw materials tends to be very low (0.06–0.7%), the cobalt content in the active mass of portable LiBs can be up to about 20 wt.%. The lithium content in ores and in portable LiBs is approximately the same (1–5%) [4].

Lithium batteries can be divided into portable, automotive and industrial batteries depending on the point of application. According to Directive 2006/66/EC, portable batteries are defined as airtight, hermetically closed cells that can be hand-carried, except for automotive and industrial batteries [5]. The new regulation will introduce a weight limit of 5 kg in order to differentiate portable batteries and accumulators from industrial ones [2]. In general, the recycling of spent lithium batteries can be divided into pre-treatment processes, pyrometallurgical processes and hydrometallurgical processes. These processes are most often combined in order to achieve comprehensive treatment, economic and environmental requirements and maximum recovery of the present components.

Among the pre-treatment procedures can be included: manual dismantling, discharge of residual voltages, mechanical-physical processing and thermal processing.

Pyrometallurgical processing methods include roasting and melting in different types of furnaces.

Hydrometallurgical processes are an alternative or a complement to another type of treatment for spent portable LiBs. The first step consists of leaching, followed by refining of the leachate and finally extraction of metals from the leachate. Spent portable LiBs enter the leaching process after mechanical-physical or thermal pre-treatment. Hydrometallurgical treatment is most often carried out on the active mass of spent portable LiBs. Active mass is a mixture of cathode and anode active material containing the main metals of interest such as cobalt; lithium; to a minor extent, nickel; manganese; and also containing significant amounts of graphite from the anode material. The active mass usually contains minor amounts of copper, aluminum and iron, which represent electrode residues and which could not be removed by available mechanical-physical methods.

For the extraction of cobalt and lithium from the active mass, acid leaching is mainly applied with the following leaching reagents: sulfuric acid, hydrochloric acid, nitric acid, various organic acids, etc., most commonly in the temperature range 25–100 °C. At the same time, nickel and manganese are mostly leached. In the case of cobalt, reductive leaching is appropriate.

Refining of the obtained leachate is carried out if the leachate has also received the accompanying metals and other impurities. The valuable metals, especially cobalt and lithium, and to a minor extent nickel and manganese, may be extracted from the solution:by precipitation in the form of sparingly soluble compounds;by cementation in the form of a cementation precipitate;by solvent extraction;by electrowinning.

Ion exchange is also an option. Electrochemical processes are another method for obtaining metals from active mass. The chosen technology is mainly based on the composition of the leachate, the content of the interested metals and the presence of accompanying metals and other impurities in the leachate.

The aim of this paper is to summarize the available information and to provide an overview of current technologies and trends in the recycling of spent portable LiBs in the research and at a practical scale, to describe the advantages and disadvantages of proposed or preferred approaches.

## 2. Characterization of Portable LiBs

Portable LiBs can be divided into primary non-rechargeable and secondary rechargeable batteries. Compared with portable zinc or alkaline batteries, portable non-rechargeable LiBs have higher specific energy, high nominal voltage, high current density, long discharge cycle, light weight, etc. They are used in various electronic applications, e.g., in medical devices such as hearing aids [6,7,8].

Primary LiBs consist of a lithium metal anode and a cathode, which can be composed of several materials, e.g., MnO_2_, SOCl_2_ and FeS_2_. Li-MnO_2_ batteries are the most common type of portable non-rechargeable LiBs. They are produced in cylindrical shapes in different sizes and as buttons or coin cells. Portable Li-SOCl_2_ batteries contain a liquid cathode, which consists of a porous carbon current collector filled with thionyl chloride [6]. Furthermore, primary LiBs contain iron, aluminum and plastics coming from the cover, organic electrolytes containing lithium salts, polypropylene-based separators, carbon as a catalyst and possibly others. The composition of portable primary LiBs of cylindrical and button shape from selected producers is given in Table 1.

Recently, primary lithium batteries often replace alkaline batteries in cameras, children’s toys, watches and other electronic devices due to their advantageous properties. Their use is mainly limited by their higher cost. The disadvantage of primary LiBs is the risk of explosion and fire in case of improper handling, dismantling and recycling, which is mainly related to the high reactivity of metallic lithium and the content of flammable organic electrolytes.

Portable rechargeable LiBs have a wider application than non-rechargeable ones and are used in mobile phones, laptops, tablets, electronic toys, etc. The advantages of portable rechargeable LiBs can be listed as long life and charging cycle, high specific energy and energy density, wide operating temperature range, no memory effect, long shelf life and more. The main disadvantages include relatively high cost, cell degradation at high temperatures, cell overcharging in improper use, which can lead to overheating and degradation of the components, followed by combustion and even explosion [7,8].

Portable secondary LiBs contain lithium, mostly in the chemical compounds. The cathode active material in portable secondary LiBs tends to be mainly LiMeO_2_-based metal oxides deposited on an aluminum foil cathode, where Me = Co, Ni, Mn and their combination. The anode active material consists of carbon or graphite deposited on a copper foil-anode. Between the anode and the cathode there is a porous separator, most often made of polypropylene, which serves as an electric insulator [8].

The most widely used cathode active material for portable secondary LiBs so far is LiCoO_2_ (LCO), due to its wide application temperature range (~10–100 °C), high specific and bulk capacitance, low self-discharge and the longest lifetime among currently available materials [13,14]. LiMn_2_O_4_ (LMO) represents a cheaper alternative to cathode active material; it is used in a variety of applications such as drills and other power tools, medical devices, e-bikes and also EVs [15,16].

Recently, the use of LiNiMnCoO_2_ (NMC) with reduced Co content compared with LCO, which partially replaces Ni and Mn, has been increasing. NMC is one of the most successful formulas for cathode material and is becoming a popular material for electric tools, e-bikes and EVs as well [17]. Furthermore, LiMePO_4_-based compounds, where Me = Fe, Co, Mn, Ni, are also used as cathode active material. LiFePO_4_ (LFP) is mainly used due to its favorable price, good thermal stability and large specific capacity. However, the disadvantage is its low conductivity [18,19,20].

The electrolytes include ethylene carbonate, propylene carbonate, dimethyl carbonate and dimethyl sulfoxide, similar to primary LiBs. Li salts are dissolved in organic solvents, e.g., LiClO_4_, LiF and LiPF_6_ [17,21]. To improve the conductivity, carbon or acetylene black is added to the cathode material [18,22]. The material composition of portable secondary LiBs is shown in Table 2.

## 3. Possibilities of Spent Portable LiB Recycling

Recycling processes include pre-treatment procedures, pyrometallurgical, hydrometallurgical and combined processes.

### 3.1. Pre-Treatment

Pre-treatment processes for spent portable LiB recycling include manual dismantling, residual voltage discharge, mechanical-physical treatment and thermal treatment to isolate and concentrate the material suitable for further processing.

Figure 1 shows a principle scheme for the pre-treatment of spent portable LiBs.

Manual dismantling is carried out in order to release and isolate the individual components. In industry, it is particularly applicable for the processing of oversized lithium cells, e.g., from EVs. The discharge of the residual voltage of spent portable LiBs may be carried out by thermal, wet or cryogenic processes, in particular to avoid short circuits and subsequent burning of the present electrolytes and binders. The aim is also to reduce the reactivity of the metallic lithium present in the primary LiBs. The mechanical-physical processing of spent portable LiBs mainly involves crushing, grinding and subsequent sorting. The output of the mechanical-physical processing is an active mass, metallic electrodes, aluminum packaging in the non-ferrous fraction, steel packaging in the ferrous fraction and separators and plastic packaging in the non-metallic fraction.

In current research, various techniques and combinations of crushing and grinding are being developed and validated in order to efficiently disintegrate the spent portable LiBs and release the maximum amount of the individual components, especially the active mass. The aim is also to avoid undesirable phenomena during crushing or grinding, e.g., encapsulation of the active mass inside the shell [23]. An example of the efficient recovery of active mass from spent portable LiBs is the mechanical-physical pretreatment procedure using a double-rotor crusher followed by a hammer crusher with a gain of active mass in the <0.71 mm fraction, which accounts for 40–57% of the input sample weight [24,25]. The active mass from portable LCO-based LiBs can contain more than 20% of Co, about 4% of Li and a minor fraction of Mn, Ni, Al, Fe and Cu (<1%). Al, Fe and Cu represent impurities in the active mass as residual metal electrodes and packaging. Graphite contributes to ca. 50% of the active mass [24].

In addition to mechanical-physical pre-treatment, thermal pre-treatment is frequently used, e.g., heating to evaporate volatile components, pyrolysis, calcination and others. Potentially hazardous gases released during thermal treatment have to be captured and cleaned in flue gas cleaning installations. The captured evaporated electrolyte can be recycled theoretically.

The aim of pyrolysis of the spent LiBs is to remove separators, binders and other organic components or changes in the structure in order to obtain material with a suitable composition for further processing. Calcination is applied for a similar purpose to pyrolysis or to modify the composition of the final product.

One way to effectively separate the cathode material from the aluminum cathode is to use a variety of organic reagents that dissolve the present binders. Organic solvents for dissolving binders are usually applied after manual dismantling or mechanical-physical treatment. Perspective organic solvents are n-methylpyrrolidone, dimethylacetamide, acetone and trifluoroacetate (TFA) [26,27,28,29,30]. A PVDF-based binder can also be dissolved using dimethylformamide; however, a polytetrafluoroethylene (PTFE)-based binder does not dissolve in the above medium [31].

The cathode active material can be separated from the Al cathode via leaching with NaOH as well [28,32] or by wet ultrasonic separation combined with crushing. Carbon can also be separated from the copper anode in this way [33,34].

### 3.2. Pyrometallurgical Processing

Pyrometallurgical treatment methods for spent portable LiBs include roasting and melting in different types of furnaces, e.g., in a shaft furnace, rotary kiln or converter.

Figure 2 shows a principle scheme for the pyrometallurgical processing of spent portable LiBs.

#### 3.2.1. Roasting

During roasting of spent portable LiBs, various insoluble metal compounds convert into products that are soluble, e.g., even in water. In this way, the use of the aggressive leaching reagents required in the conventional hydrometallurgical process is eliminated. Prior to roasting, it is necessary to carry out a mechanical-physical pre-treatment. Depending on the used reagent, the roasting processes can be divided into chlorination roasting, sulfating roasting, nitration roasting, reduction roasting and possibly others. Chlorination roasting involves heating the feed material together with a chlorinating agent such as HCl_(g)_, NH_4_Cl, NaCl or Cl_2(g)_ to produce readily soluble chlorides of the cobalt and lithium [35]. In sulfating roasting, the following sulfating reagents—SO_2(g)_, MgSO_4_, NH_4_SO_4_, NaHSO_4_.H_2_O and Na_2_SO_4_—have been confirmed to produce readily soluble Li_2_SO_4_ [36].

In reduction roasting, the active mass is heated with a reducing agent, which can be carbon, charcoal or coke. The carbon or graphite present in spent LiBs is often used for this purpose [37]. New phases such as Li_2_CO_3_, CoO and NiO are formed during roasting at temperatures of 600–900 °C. However, the use of higher temperatures (ca. 700 °C) results in graphite burning out, thus reducing the reduction conditions [38]. The use of a vacuum at higher temperatures (ca. 900 °C) is preferred, where the carbothermal reduction of the present oxides to the metallic form occurs. The carbothermal reduction during roasting of the cathode material containing LiCoO_2_, the decomposition of LiCoO_2_ and the formation of Li_2_CO_3_ proceeds according to reactions (1)–(3) [39].
LiCoO_2_ + 3C = Li_2_CO_3_ + 4Co + CO_2_,(1)
LiCoO_2_ = Li_2_O + CoO + O_2_,(2)
Li_2_O + CO_2_ = Li_2_CO_3_.(3)

Waste biomass can be used as an alternative reducing agent for the carbothermal reduction of metals during roasting [40].

#### 3.2.2. Melting

The melting of spent portable LiBs consists of two steps. First, they are heated to a lower temperature in a furnace in order to discharge residual voltages and evaporate the electrolyte. In the second step, metal alloys are formed at high temperatures in a reducing carbon atmosphere and other organic materials such as plastics and separators are burned off. Carbon from the anode and aluminum from the cathode act as reducing agents. In the pyrometallurgical process of spent LiBs, cobalt and nickel from the cathode active material and copper from the anode are recovered most efficiently. The obtained alloy usually proceeds to hydrometallurgical processing. Lithium passes to the slag, from where it can be recovered with additional processing [41].

During melting of spent LiBs, the high viscosity of the slag and high melting temperature are often a problem, which causes metal particles to drift into the slag during the casting process, resulting in losses. An essential tool for reducing the losses and for the efficient transfer of lithium and aluminum into the slag is an optimal slag system and multi-stage slag casting. CaO and SiO_2_ have proven to be useful as slag-forming additives. Al_2_O_3_, which comes from spent portable LiBs as an oxidized cathode, is also a suitable slag-forming additive. The resulting alloy mainly consists of a solid solution of Fe-Co-Cu-Ni, including a small amount of matte [42].

### 3.3. Hydrometallurgical Processing

Hydrometallurgical processing is another alternative or complement to the treatment of spent portable LiBs. They enter the leaching process after mechanical-physical or even thermal pre-treatment. Hydrometallurgical treatment of spent portable LiBs is most often carried out on the active mass. In the case of leaching of the active mass to recover cobalt, acid leaching with the addition of a reducing agent predominates. The interesting metals, which are mainly cobalt and lithium and, to a minor extent, nickel and manganese, can be extracted from solution via precipitation, cementation, solvent extraction, electrowinning, ion exchange and their combination.

The type of technology is based mainly on the composition of the leachate, the content of the suitable metals and the presence of accompanying metals and other impurities in the leachate.

The principle scheme for hydrometallurgical processing of spent portable LiBs is shown in Figure 3.

#### 3.3.1. Leaching

In the research of leaching valuable metal from active mass or cathode active material, various leaching systems are studied, searching for optimal parameters (temperature, concentration of reagents, L:S ratio, leaching time, etc.) in order to achieve the maximum extraction of the present metals and to explain the theoretical basis of the processes. Sulfuric acid, as the most common leaching agent in terms of availability and cost, can be considered as a suitable leaching agent for leaching the metals from the active mass of spent LiBs.

For efficient leaching of the metals, hydrogen peroxide is often added to the sulfuric acid in varying volumes.

During leaching of a LiCoO_2_-based active mass in sulfuric acid without and with the addition of hydrogen peroxide, the following reactions can be expected (4) and (5) [24,43]
LiCoO_2_ + 1.5 H_2_SO_4_ = CoSO_4_ + 0.5 Li_2_SO_4_ + 0.25 O_2_ + 1.5 H_2_O, ΔG^0^_293_= −612.2 kJ,(4)
LiCoO_2_ + 1.5 H_2_SO_4_ + 1.5 H_2_O_2_ = CoSO_4_ + 0.5 Li_2_SO_4_ + O_2_ + 3 H_2_O, ΔG^0^_293_= −617.3 kJ.(5)

In both cases, the value of ΔG^0^_293_ is negative, indicating that the progress of both reactions is thermodynamically possible. Hydrogen peroxide acts as a reducing agent to reduce Co^3+^ from the LiCoO_2_ structure to Co^2+^, which is not so resistant to leaching [43].

Using sulfuric acid without the addition of hydrogen peroxide, maximum cobalt yields of 50–60% can be achieved from the active mass of the spent portable LiBs even at higher acid concentrations and at high temperatures (80–100 °C). In the H_2_SO_4_ + H_2_O_2_, cobalt yields can reach up to 100%, depending on the concentration of the leaching agent, the amount of addition of reducing agent, the temperature, the slurry density, the leaching time, etc. In addition, lithium (95–100%) and other metals present, such as nickel, manganese, copper, aluminum and others, are also leached [44,45,46,47,48,49].

Sulfuric acid with hydrogen peroxide can also be used for metal leaching from the reduction-roasted active mass of spent LiBs with high efficiency [38]. However, the addition of hydrogen peroxide has some limitations. The problem is leaching at higher temperatures (80–100 °C), where cobalt yields decrease [50]. The decrease in cobalt yield with increasing temperature may be due to the decomposition of hydrogen peroxide, which theoretically proceeds according to reaction (6) [4,50].
2H_2_O_2_ = O_2_ + 2H_2_O, ΔG^0^_293_ = −116.41 kJ.(6)

For these reasons, a temperature between 40 °C and 60 °C can be considered as the optimum temperature for leaching Co with sulfuric acid and hydrogen peroxide.

Another reducing agent that has confirmed its potential is metallic copper, e.g., in the ratio Cu: LiCoO_2_ = 1:1. Copper also serves as a catalyst. It is also suitable to add chloride ions to this system to stabilize Cu^+^ in the form of complexes. The dissolution of Cu occurs through an intermediate step in which solid CuCl precipitates on the Cu surface, potentially inhibiting the leaching process of the valuable metals [51].

Iron scrap can also be considered as a suitable, economical and efficient reducing agent for leaching Co, Ni and Li from the active mass in sulfuric acid. The reduction is provided by the release of Fe^2+^ from its dissolution. The use of ferrous scrap significantly increases the yields of cobalt and nickel, while it has a negligible effect on the leaching of lithium. The added iron is subsequently most efficiently removed from the leachate via precipitation at pH = 4, without co-precipitation of other metals. After iron removal, cobalt and nickel can be recovered from the leachate via cementation with zinc scrap [52].

Sulfuric acid can also be used as a leaching agent for leaching lithium slag from the melting of spent LiBs. However, in addition to lithium, the use of sulfuric acid also results in the simultaneous leaching of silica, which causes the formation of gels, making it difficult to filter and recover the lithium from the leachate. An alternative in slag processing is dry digestion using concentrated sulfuric acid, which avoids the formation of gels while maintaining a high lithium yield [53].

A promising approach for the extraction of metals from the active mass of spent LiBs is the leaching and simultaneous precipitation of the present metals using oxalic acid. Under suitable conditions (e.g., 0.25 M oxalic acid, 10% slurry density, addition of 0.5 vol.% H_2_O_2_, temperature 80 °C, leaching time 90 min.), 100% of copper and lithium can be selectively converted to a leachate. At the same time, oxalates of cobalt, nickel and manganese are formed. Diffusion of oxalic acid to LiCoO_2_ occurs through a layer of solid reaction product which forms on the surface; this step is probably rate limiting. The formed oxalates can be subsequently leached in, e.g., sulfuric acid or directly used as precursors for the synthesis of new cathode material [54,55,56].

Another suitable leaching agent for metals extraction from the active mass of the spent LiBs is hydrochloric acid [43,57,58,59]. The leaching of Co and Li from LCO-based active mass using HCl is represented by the following chemical reaction (7) [43]:2LiCoO_2_ + 8HCl = 2LiCl + 2CoCl_2_ + 2H_2_O + Cl_2_, ΔG^0^_293_ = −337.16 kJ.(7)

Practically, almost 100% of Co and Li can be leached from the active mass using 2–4 M HCl at 60–80 °C in a relatively short time (0.5–1 h) [24,58,59]. A comparison of the effect of H_2_SO_4_ and HCl shows that maximum yields of Co and Li can be achieved using HCl without the addition of any reducing agents, whereas a reducing agent is required for 100% leaching of the valuable metals using sulfuric acid. The cobalt extraction in both H_2_SO_4_ and HCl takes place in two time periods. In the case of the leaching in H_2_SO_4_, in the first period, i.e., 15–20 min from the start of leaching, the process is controlled by the rate of the chemical reaction, Ea_(Co)_ = 43–48 kJ·mol^−1^. In the second time period, the process changes to a diffusion-controlled rate, as demonstrated by the apparent activation energy, Ea_(Co)_ = 3–3.5 kJ·mol^−1^. In the case of hydrochloric acid, the Co extraction in the first time period is controlled by the rate of the chemical reaction, Ea_(Co)_ = 40–44 kJ·mol^−1^. In the second time period, the process has a mixed mechanism, Ea_(Co)_ = 20–26 kJ·mol^−1^. The lithium extraction is diffusion controlled in both time periods, or proceeds in a mixed mode, Ea_(Li)_ = 2–20 kJ·mol^−1^, in both leaching agents. The extraction of cobalt and lithium is influenced by the internal structure of the LCO-based active mass, and the extraction of cobalt depends on the extraction of lithium from the LiCoO_2_ structure [24].

Other leaching reagents such as Na_2_CO_3_, ammonia reagents and H_3_PO_4_ are also suitable for leaching the active mass from the spent LiBs. Na_2_CO_3_ has proven to be a suitable leaching agent for the leaching of nearly 99% of Li from the roasted active mass of the spent LiBs. Ammonia leaching of roasted active mass (e.g., NH_3_.H_2_O + (NH_4_)_2_CO_3_ + Na_2_SO_3_ leaching system) can leach metals such as cobalt and nickel in nearly 100% yields after lithium leaching, leaving manganese in the insoluble residue [60]. Ammonia leaching can also be carried out at higher pressure using various reducing agents—Na_2_S_2_O_3_, Na_2_HPO_3_ and (NH_4_)_2_SO_3_. Using (NH_4_)_2_SO_3_, up to 100% recovery of Co, 98% recovery of Ni and more than 90% recovery of Li can be achieved [61]. H_3_PO_4_ with hydrogen peroxide can also provide a suitable leaching medium for Co and Li leaching from the active mass at high temperatures (90 °C) [62].

Among the intensification methods for metals extraction from the active mass of the spent LiBs can be included ultrasonic leaching, which aims to achieve high metal yields in a short time. Ultrasonic leaching (360 W) of the cathode active material using H_2_SO_4_ with H_2_O_2_ can achieve more than 90% of Co and Li yields in a short time of 30 min. Ultrasonic leaching provides a higher leaching rate and increased process efficiency compared with conventional leaching under otherwise identical experimental conditions [63].

In recent years, several organic acids such as citric acid [64,65], tartaric acid [66], lactic acid [67], adipic acid [68] and DL-malic acid [69] have been studied as leaching agents for Co and Li extraction in the research. The use of organic reducing agents, e.g., ascorbic acid [70,71], glucose and fructose [57,72] is also being investigated. The advantage of using reducing saccharides as reducing agents is that wastes from the sugar industry or agriculture can be used as saccharide sources [57].

The use of such leaching systems is mainly aimed at achieving higher process selectivity, lower toxicity and lower environmental impact. The disadvantage is the high cost compared with conventional mineral acids. High metal yields can be achieved in these systems, but the disadvantage, for instance in the case of adipic acid and glycine, is the long leaching times (3–6 h) [69,73].

In addition to the interesting metals, a significant part of the active matter consists of graphite, which remains in the insoluble residue during leaching. There are also studies describing the possibility of recovering graphite from the anode during the processing of LiBs. The most effective seems to be its separation during leaching, when it is concentrated in the insoluble residue [74].

#### 3.3.2. Refining and Recovery of Metals from Leachates

The leachates from spent portable LiBs intended for further processing mainly contain cobalt—approximately five times more cobalt than lithium, magnesium and nickel. The other accompanying metals (iron, aluminum, copper) are only present in minor concentrations. The processes used to extract the valuable metals from the leachate include cementation, solvent extraction, precipitation, electrochemical processes, ion exchange and combinations of these.

Cementation

During the cementation of valuable metals, Al, Zn and Fe can be used as a cementing agent [75]. In practice, zinc powder is most commonly used. The disadvantage is that the obtained cementation precipitate is a mixture of the several present metals. In addition to cobalt, the present nickel, copper and other metals are also cemented. Subsequently, such a cementation precipitate has to be refined. In contrast, cementation as a method for recovering metals from leachates cannot be used for recovering lithium because of its high electronegativity, which also provides the possibility of mutual separation of, e.g., cobalt and lithium.

Precipitation

Using precipitation to produce sparingly/slightly soluble or insoluble compounds, almost all present metals can be recovered from the leachate after leaching of the active mass of spent LiBs, depending on the precipitating agent and other parameters. Examples of compounds of the valuable metals, the Ksp values and their solubility in water at 25 °C are given in Table 3.

The difficulty with precipitation is that, in addition to the required metal, other present metals also pass into the precipitate. High process selectivity can be achieved via multi-step precipitation with an appropriate choice of precipitating reagents and a suitably chosen pH. Cobalt can be efficiently precipitated from sulfate leachates after leaching of the active mass by several proven reagents such as NaOH, KOH, Na_2_CO_3_, oxalic acid and ammonium oxalate [79,80,81]. The behavior of the metals during precipitation can be predicted through speciation diagrams [44].

Oxalic acid has been confirmed as a suitable precipitating agent for cobalt, but it also precipitates copper, nickel and possibly other metals [79]. Direct use of oxalic acid is appropriate if the leachate contains no or minimal amounts of copper [82]. If copper is present in the leachate, it is preferable to precipitate it first. Sodium sulfide is suitable for precipitating copper with high efficiency (up to 99.9%) to form copper sulfide at a molar ratio of Na_2_S:Cu^2+^ = 3:1, at 25 °C, with a precipitation time of 30 min [44].

In the case of a targeted preparation of a mixed precipitate, oxalic acid may be the sole used reagent. The advantage in such a recycling process is the absence of complicated metal separation and low processing costs. The mixed precipitate of cobalt, manganese and nickel oxalates can serve as a precursor in the production of new cathode materials [83].

Dimethylglyoxime may be considered as a suitable nickel precipitating agent, and nickel precipitation may be included at the beginning of the process or preceded by manganese precipitation using KMnO_4_. Nickel precipitation by dimethylglyoxime is represented by chemical reaction (8) [84,85,86]
Ni^2+^ + 2C_4_H_8_N_2_O_2_ + 2OH^−^ = [Ni-(C_4_H_7_N_2_O_2_)_2_] + 2H_2_O.(8)

Nickel and manganese can also be effectively precipitated using Na_2_CO_3_ at a suitable pH. Manganese is selectively precipitated as MnCO_3_ at pH = 7.5 and nickel as NiCO_3_ at pH = 9. At the highest pH = 12–14, lithium is precipitated as Li_2_CO_3_ by Na_2_CO_3_, but often only after the solution has been concentrated to the required Li concentration (ca. 10 g·L^−1^) [81]. Lithium is most often precipitated from the leachate as a last step. With a molar ratio of Li:Na_2_CO_3_ = 0.7, pH = 12 and temperature = 100 °C, a lithium precipitation efficiency of almost 99% can be achieved in a time of 40 min [84]. Na_3_PO_4_ is also appropriate for lithium precipitation from the acid leachate with high efficiency (more than 90%), producing Li_3_PO_4_ [87].

Solvent extraction

In addition to cementation and precipitation, solvent extraction (SX) can be used in the recovery of metals from these types of leachates using acid extraction reagents [88,89,90]. Solvent extraction is characterized by several parameters, such as distribution ratio, partition coefficient, separation factor and extraction efficiency. In particular, the extraction efficiency and selectivity of the process are influenced by the type of organic extraction reagent and its concentration, the type of stripping reagent and its concentration, the pH, the extraction time, the ratio of the A:O = inorganic (aqueous) phase to the organic phase during extraction, the ratio of the O:A = organic phase to the aqueous phase during stripping and possibly others. Operating pressure and temperature have negligible or only a small influence [91]. Among the reagents used in the SX of valuable metals can be included Cyanex 272, D2EHPA, ALIQUAT 336, VERSATIC, TBP—tributyl phosphate, etc.

Cyanex 272 has proven to be an excellent extractant for the mutual separation of cobalt and nickel, where the cobalt passes into the organic phase with high efficiency and the nickel remains in the leachate [92]. The cobalt extraction process using Cyanex 272 can be written by Equation (9) [92].
Co^2+(aq)^ + (HA)_2(org)_ ⇔ CoA_2(org)_ + 2H^+(aq)^.(9)

From the sulfate leachate, 100% of Co extraction efficiency can be achieved at pH = 5.5–8; however, other metals present—Ni and Mn—can be co-extracted along with Co. The optimum pH can therefore be considered to be pH = 5.5–7, where 90–100% of Co passes into the organic phase and nickel remains in the leachate.

If manganese is also passed into the enriched solution, it is advisable to pre-treat the cobalt recovery with selective manganese extraction, e.g., with D2EHPA at pH = 4 [25,92]. The extraction reagent D2EHPA at a concentration of 12% *v*/*v* is suitable for the extraction of copper (90–100%) and aluminum (80%) at pH = 1.5–3, depending on the type of leachate, in addition to the extraction of manganese [64,87]. At higher reagent concentrations (about 20 vol.%), it can also be used for the extraction of lithium at pH = 5.5, O:A = 4 with 94% efficiency [64].

Other reagents for the selective extraction of cobalt include Mextral^®^272P, using which Co is extracted but nickel and lithium remain in the leachate [93]. The use of a two-stage countercurrent system is advantageous, where nearly 100% of cobalt can be obtained under certain conditions [84,94]. If copper is present in the leachate, it can be selectively extracted using Mextral^®^5640H with efficiencies up to 100% at pH~2 using two countercurrent extraction stages [93]. LIX 84-IC can also be used to extract copper at pH = 3–6, while nickel is extracted as well [95]. Another useful extraction reagent for the extraction of cobalt from leachates after acid leaching is PC-88A. Together with cobalt, manganese is co-extracted at pH = 5, but Ni and Li remain in the leachate [96].

Recently, the use of ionic liquids Cyphos IL-102, Cyphos IL-101 [97,98] and deep eutectic solvents (DESs) [99] in various combinations has also been investigated for the solvent extraction of cobalt. Cyphos IL-102 has confirmed its potential for the selective quantitative extraction of cobalt at a concentration of 0.2 M using countercurrent extraction at a phase ratio of A:O = 1, with metals such as Mn and Li not being extracted [97]. Similarly, Cyphos IL-101 can be described as a suitable extraction reagent for cobalt with excellent stability and a high extraction capacity of about 35 g·L^−1^ [98]. Among the DESs group, the combination of choline chloride with phenylacetic acid in a 1:2 ratio can be considered among the most effective [99].

Ion exchange

Ion exchange rarely occurs in the leachate treatment process after leaching of the active mass of spent LiBs. It is generally more suitable for dilute solutions, e.g., wastewater, where metal concentrations are much lower than those in leachates. Potentially suitable resins for cobalt recovery include a chelating polyamide oxime resin, which has confirmed its potential in recovering cobalt from the leachate after the leaching of LCO-based cathode material in HCl. High adsorption of cobalt (97%) can be achieved at pH = 5.5 with a resin content of 0.04 g·mL^−1^ after 15 min of processing. The advantage of using this resin is that alkali metals and rare earth metals do not form complexes, that is, lithium remains in the leachate [100]. Dowex M4195 can also be described as an resin suitable for the selective recovery of cobalt and nickel from leachates, with an efficiency of almost 99% for both metals [101].

#### 3.3.3. Electrochemical Methods

Electrowinning as a method for metals’ recovery from leachate after leaching of the active mass of spent LiBs can be effectively used for the recovery of cobalt and nickel or manganese. However, refined solutions with a high content of the valuable metal have to be fed to the process. The problem in the electrowinning of cobalt is the presence of nickel. The small difference in the standard E^0^ potentials of these metals (E^0^_Co2+/Co_ = −0.28 V, E^0^_Ni2+/Ni_ = −0.25 V) makes it almost impossible to selectively electrolytically recover them when they are in solution together. For this reason, it is preferable if the electrowinning is preceded by their mutual separation, e.g., by solvent extraction. Cobalt electrowinning is also sensitive to the content of other metals such as Fe, Cr, Cu, Pb, Zn, etc., because cobalt has a negative electrochemical potential, which means that metals with a positive potential are preferentially deposited at the cathode [102].

Among the perspective electrochemical procedures can be included the recycling of LCO-active masses via a neutral aqueous electrolytic cell. During this process, an electrochemical pH gradient is generated via the electrolysis of water, which allows the leaching of metals at low pH in the anode unit and the precipitation of solid Co(OH)_2_ at high pH in the cathode unit. In addition, lithium in the form of Li_2_CO_3_ is obtained [103]. The above method can also be used in the recycling of LMO-based active mass [104]. The LCO-active mass can also be electrochemically treated by simultaneous leaching and metal precipitation from the solution. In the first step, the metals are leached using H_2_SO_4_, which is formed in the reactor during the electrodeposition of copper from the copper sulfate electrolyte. Extracted Co^2+^ ions are recovered electrolytically as metallic cobalt with simultaneous regeneration of sulfuric acid. Lithium is recovered from the leachate as Li_2_CO_3_ [105].

## 4. Industrial Processing of Spent Portable LiBs

In the industry, pyrometallurgical and combined processing of spent portable LiBs is currently predominant. In combined processing, the pyrometallurgical step is usually followed by a hydrometallurgical refining step. The purpose of these hydrometallurgical processes is the efficient separation of the valuable metal from the mixed material as an alloy, a matte or a slag—and its extraction in the form of commercially interesting compounds. These are mostly “in-house” developed methods, where the recycling procedures are developed by the companies themselves [106,107,108,109,110]. A general scheme for the processing of spent portable LiBs in the industrial sector is shown in Figure 4.

### 4.1. Accurec Recycling

Accurec Recycling is a German company that deals with the recycling of almost all types of spent LiBs. The processing procedure consists of three basic steps—residual voltage discharge, thermal pre-treatment and sorting. Thermal pre-treatment is carried out in a rotary kiln where the temperature does not exceed 600 °C in order to avoid oxidation of metals such as aluminum. Subsequently, the cells are crushed and the obtained fractions are multi-stage sorted using magnetic separation and air sorting. The output of this process is an iron fraction, a Cu/Al fraction and a Co-Ni active mass. The metal fractions are sold for further processing. From an active mass with a particle size < 0.2 mm, pellets are produced. The pellets are treated with a carbothermal reduction at 800 °C to produce a cobalt alloy. The lithium is passed into the dust and slag. Lithium from slag can be recovered by subsequent hydrometallurgical steps [109,110]. A flowsheet of the described process is shown in Figure 5.

### 4.2. Akkuser

AkkuSer in Finland is involved in the mechanical and physical pre-treatment of several types of spent electrochemical cells using dry technology to obtain enriched fractions for further processing. The pre-treatment process for sorted spent LiBs consists of crushing using two types of crushers. The first crusher operates at a temperature of 40–50 °C, at 100–400 rpm, reducing the cell size to 1.25–2.5 cm. The crushed material is then transferred to the secondary crusher, which operates at 1000–1200 rpm. This crusher reduces the material to a size of <6 mm. Iron is removed from this fraction via magnetic separation. The resulting fraction with Co-Cu content is a saleable product [110,111]. A flowsheet of AkkuSer’s spent LiB recycling process is shown in Figure 6.

### 4.3. Batrec AG

Batrec AG’s facility in Switzerland processes all types of electrochemical cells [110]. At the company, spent LiBs are subjected to mechanical pre-treatment by crushing them in a protective CO_2_ atmosphere to minimize possible reactions. Subsequently, the cells are exposed to humid air. The fractions obtained by the crushing and subsequent sorting are sold for further processing [7]. A flowsheet of the described process is shown in Figure 7.

### 4.4. Glencore

Canadian Glencore, which previously operated as XSTRATA, uses a combined processing of spent LiBs. Spent LiBs are injected to a rotary kiln or the material is melted in a converter. The temperature in the melt reaches 1300 °C. A formed matte is subsequently processed by a hydrometallurgical route in order to complete cobalt extraction. The organic components are burned in the melting process and the resulting gas is treated via additional combustion. The disadvantage is the loss of lithium in the slag [106,110,112]. A flowsheet of Glencore’s recycling process is shown in Figure 8.

### 4.5. Inmetco

At the American company INMETCO, spent LiBs represent a secondary input to a pyrometallurgical process that was originally designed for steelmaking waste treatment. Recycling of spent LiBs involves mechanical pre-treatment and reduction melting. Pre-treatment consists of disassembly, electrolyte evaporation and subsequent crushing. Crushed lithium cells are melted together with the pellets from Ni/Cd waste. The process is carried out at 1260 °C for 20 min. The organic components and the present carbon are combusted or used as a reducing agent in the carbothermal reduction of the metals. A submerged electric arc furnace is used to refine the melt, yielding Co, Ni and Fe in alloy form, and lithium is passed to the slag [109,110]. A flowsheet of the recycling process is shown in Figure 9.

### 4.6. JX Nippon Mining and Metals

Japanese company JX Nippon Mining and Metals operates a pyro-hydrometallurgical method for spent LiB recycling. The spent LiBs are injected into a stationary furnace, where they are subjected to thermal pre-treatment in order to obtain the cathode material and the other usable components separately. The organic electrolyte is evaporated and the evaporated fluorine is recovered via precipitation. The material obtained from the thermal pre-treatment is crushed and subsequently sieved and separated into several grain size fractions. The fine fraction, which mainly consists of cathode material, proceeds to leaching. The metals are recovered from the leachate via multiple solvent extraction followed by precipitation in the case of Mn recovery and electrowinning in the case of Co and Ni recovery. The main products of the process are electrolytic Co and Ni. The by-products are Li_2_CO_3_ and MnCO_3_ [113]. A flowsheet of the described process is shown in Figure 10.

### 4.7. Recupyl Valibat

The French company Recupyl treats spent LiBs in the Valibat recycling process. At first, spent LiBs are subjected to a two-stage crushing process. The first stage of crushing takes place in a rotary knife crusher in an argon or CO_2_ atmosphere, where the metallic Li present in the non-rechargeable LiBs is passivated to Li_2_CO_3_. In the second stage of crushing, an impact crusher is used at 90 rpm in order to reduce the particle size below 3 mm. The material is then separated into two fractions using a vibrating screen. The oversized fraction is subjected to magnetic separation to remove the iron present. The non-magnetic fraction is then processed on a densimetric bench separator. This results in a high density fraction containing Cu and Al and a low density fraction consisting of paper and plastics. The fraction <3 mm is further sorted using 500 μm sieves, where most of the Cu is removed. The fine fraction <500 μm, rich in metal oxides, is then mixed with water and pH is adjusted to 12. The lithium salts present are dissolved and the metal oxides and graphite are concentrated in the insoluble residue. Lithium is recovered from the solution by the addition of CO_2_. The insoluble residue is leached with H_2_SO_4_ at 80 °C in order to obtain Co. Residual copper is recovered from leachate via cementation. Any remaining lithium may be precipitated as Li_3_PO_4_ at this point by the addition of H_3_PO_4_. Finally, cobalt is precipitated from a solution with NaClO as Co(OH)_2_, or is obtained electrolytically [114,115,116]. A flowsheet of the described recycling process is shown in Figure 11.

### 4.8. Retriev Technologies

Retriev Technologies’ forerunner was TOXCO, which processed spent portable non-rechargeable LiBs. Today, the Canadian company also recycles spent rechargeable LiBs. The process involves spent LiB discharging with a brine solution and crushing. If spent non-rechargeable LiBs are processed, cryogenic crushing is used. The crushed cells are passed in a hammer mill and then the metallic fractions are removed by sieving. Active mass is removed from the aluminum and plastic fractions on a vibrating table. Active mass is leached in water, concentrating the lithium in the leachate. Lithium from solution is obtained with the Na_2_CO_3_ as Li_2_CO_3_. Retriev Technologies also has a patented process for obtaining high quality cathode active material [110,112,117]. A flowsheet of the described recycling process is shown in Figure 12.

### 4.9. SNAM

French company SNAM focuses on recycling almost all types of electrochemical cells, including spent portable and industrial LiBs. In the first step, spent LiBs are subjected to a discharge process. This is followed by pyrolysis in order to remove separators and electrolytes. Subsequent sieving produces fractions containing Fe, Al, Cu and a fine fraction, which is processed hydrometallurgically with cobalt and lithium, obtaining saleable products [118,119]. A flowsheet of the described recycling process is shown in Figure 13.

### 4.10. Sumitomo Metal Mining

Sumitomo Metal Mining (SMM) uses a combined method of recycling spent LiBs. It is also designed to process spent LiBs from EVs. The smelting process produces a Cu-Co-Ni alloy and Li slag. Both intermediates are processed using hydrometallurgical refining. The output of the process is a new cathode active material and electrolytic copper [120,121]. A flowsheet of the described recycling process is shown in Figure 14.

### 4.11. Umicore Battery Recycling

Umicore Battery Recycling in Belgium processes all types of spent LiBs—from portable to industrial—using reduction melting in a vertical shaft furnace [122]. The feed consisting of spent LiBs, NiMH batteries and slag-forming additives—SiO_2_, CaCO_3_ and coke—is supplied to the preheating zone of the shaft furnace with a temperature of up to 300 °C, where the present electrolyte is evaporated. The feed is then passed to the pyrolysis zone of the furnace where, at a temperature of 700 °C, all the organic components are completely decomposed, which also serves as fuel. This partially reduces the required coke content. The melting of the input material begins in the third zone of the furnace, where temperatures of 1200 to 1450 °C can be reached by injecting oxygen. The melting process results in three intermediate products: a Cu-Fe-Ni-Co alloy, slag and blast furnace gas. The alloy proceeds to leaching with HCl and the subsequent extraction of metals from the leachate via solvent extraction. The obtained products are Cu, Fe, CoCl_2_ and Ni(OH)_2_. The lithium is passed to the slag, where it is recovered by a complex hydrometallurgical process. Subsequently, LiCoO_2_ is produced via synthesis with CoCl_2_ [122,123,124]. A flowsheet of the described recycling process is shown in Figure 15.

### 4.12. Pilot Plant

#### 4.12.1. Battery Resources Recycling Process

The process called the Battery Resources Recycling Process (BRRP) is a combination of mechanical and hydrometallurgical processing of spent LiBs using a thermal step in the refining stage. The process consists of hammer crushing followed by magnetic separation. The non-magnetic fraction is leached in NaOH to extract Al as NaAlO_2_. The insoluble residue is separated into two fractions by sieving. The oversized fraction is sorted on the basis of different densities and a copper-rich fraction is obtained. The fine fraction is passed to a four-stage hydrometallurgical process. In the first stage, the material is leached in H_2_SO_4_ with the addition of H_2_O_2_ at 65–70 °C. The undissolved graphite is separated and the lithium is recovered from solution as LiFePO_4_. The leachate containing Co, Ni, Mn, Li, Al and Cu is subjected to pH adjustment to pH = 6.5 by the addition of NaOH in order to precipitate the remaining Al. Gaseous N_2_ is also added at this stage to limit the oxidation of Mn^2+^ ions. In the third leaching stage, after adjusting the leach composition with pure compounds to achieve the required Co:Ni:Mn = 1:1:1 ratio, the pH of the solution is raised to 11 with NaOH, resulting in the precipitation of Co, Mn and Ni hydroxides. In the last step, Na_2_CO_3_ is added to the leachate at 40 °C to precipitate the remaining lithium. From the obtained Co(OH)_2_, Mn(OH)_2_ and Ni(OH)_2_ together with precipitated and pure Li_2_CO_3_, the pellets are prepared for the production of a new cathode material [110,124]. A flowsheet of the described recycling process is shown in Figure 16.

#### 4.12.2. LithoRec

The LithoRec process focuses on obtaining high-quality cathode material, mainly by processing spent LiBs from EVs. These cells have a higher content of plastics compared with portable LiBs, which also require a more complex recycling process, especially in the pre-treatment area. LithoRec focuses on mechanical-physical pretreatment and subsequent hydrometallurgical operations. This process produces metals such as Co, Ni and Mn in oxide form and lithium in the form of LiOH and/or Li_2_CO_3_. The obtained intermediates are processed via calcination to form a precursor material suitable for the production of a new cathode material [7,110,123,125]. A flowsheet of the process is shown in Figure 17.

#### 4.12.3. OnTo Process

Additionally, the OnTo pilot process focuses on spent LiBs from EVs. Discharging is followed by disassembly and the removal of oversized components from the input material. The treated cells proceed to a perforation. Then, they are exposed to CO_2_ under supercritical conditions (pressure 7.4 MPa and temperature 31 °C). These conditions are designed to allow extraction and recovery of the organic solvent and electrolyte. The remaining cell is stabilized at atmospheric pressure and temperature. This is followed by crushing and sorting. The obtained fractions are refined hydrometallurgically [110,126]. A flowsheet of the described recycling process is shown in Figure 18.

An overview of the companies and their technologies applied in the recycling of spent LiBs is given in Table 4.

## 5. Discussion

### 5.1. Pre-Treatment

Pretreatment is an essential part of effective recycling of spent LiBs. An important step is the discharging; in industrial applications, it is frequently carried out via heating or with brine solutions (companies). The aim is to prevent burning and explosion. Manual dismantling as the subsequent pre-treatment step has been successfully applied in the industry, mainly for LiBs from EVs (e.g., Retriev Technologies, INMETCO, pilot plant LithoRec, OnTO company). It is not applicable for portable LiBs, mainly due to the small size of the cells.

The mechanical-physical pre-treatment of spent portable LiBs is a key to efficiently obtaining the individual components and fractions that can be successfully recycled in the own recycling process. Several mechanical pre-treatment processes have proven their suitability to obtain active mass, packaging materials, metal electrodes and other fractions with high efficiency and purity. Using appropriately selected steps, a major proportion of the active mass can be released and obtained with high purity; this means with a minimum of impurities such as residual electrodes, separators and packaging. There are a number of plants in the industry that only deal with mechanical-physical recycling (Batreg AG, Akkuser, Acurec) and sell the recovered fractions for further use.

The most common practices for the mechanical-physical treatment of portable LiBs include crushing and grinding, which are essential for the release of active mass. In order to make the process more efficient, multiple crushing is preferable, which is also used in industry (e.g., Akkuser, JX Nippon Minning and Metals, Recupyl Valibat). Then, sorting techniques are applied, often magnetic separation, where iron is removed from the cover, air sorting, vibrating sorting, sieving, etc. In this way, it is possible to separately recover the iron from the casing, the aluminum from the cathode, the copper from the anode and the metal oxides in the active mass. This is a sophisticated method of separating individual metals, with only minimal metal losses compared with those of pyrometallurgy, where the input tends to be whole cells without pre-treatment.

### 5.2. Pyrometallurgical Treatment

Currently, pyrometallurgical or combined processing of spent LiBs is the technology that is most widely used in the industry. The resulting metal alloy, or matte from melting, proceeds to hydrometallurgical processing. The main advantages of pyrometallurgical processes are as follows:high input variability; spent LiBs can be processed with other types of lithium cells, such as EVs batteries, as well as with other types of electrochemical cells, e.g., NiMH batteries;the relative simplicity of the process and the fact that no discharge and no mechanical-physical pre-treatment is required; spent portable LiBs can be embedded as a whole in the furnace, e.g., at Umicore Battery Recycling. Only proportionally larger pieces of industrial LiBs are dismantled;the possible use of existing plants for the treatment of other feedstock and/or wastes;the use of carbon from the anode active material, plastic packaging and separators as a partial substitute for fuel, thus improving the energy balance; in addition, the carbon also serves as a reducing agent;high capacities.

However, there are more disadvantages, mainly:
high investment and operating costs;high energy consumption;risk of explosion;the necessity of refining the obtained products;the formation of CO_2_ and other gases and their necessary capture and purification;loss of metals and other materials in the process; for example, aluminum and lithium are transferred to the slag. Currently, there are also technologies for lithium recovery from slag, but this increases the economic costs of the recycling process.

The graphite from the anode burns off in the melting process, which improves the energy balance. On the other hand, its material potential is lost, which can be considered as a disadvantage.

Research in this field is mainly focused on the optimization of the slag system in order to avoid metal losses during slag casting and for effective slagging of lithium. A promising process being investigated is roasting using temperatures up to 900 °C to suitably modify the material composition for further processing. The disadvantages of roasting are the energy consumption, the addition of chemical reagents and their cost.

A separate area is so-called direct recycling, which is a new and efficient concept for recycling spent LiBs, especially from EVs. The principle is efficient direct recycling based on targeted healing. Specifically, in the case of recycling LiFePO_4_ cathode material, this involves combining low-temperature aqueous solution relithiation and rapid post-annealing. In this way, ready-to-use recycled cathode materials can be obtained that match the electrochemical properties of the new materials. This method can significantly reduce energy consumption and greenhouse gas emissions, leading to significant economic and environmental advantages compared with current hydrometallurgical and pyrometallurgical methods [128,129].

### 5.3. Hydrometallurgical Treatment

Compared with pyrometallurgy, hydrometallurgical processes offer several advantages, for example:lower investment and operating costs;low production of gaseous emissions;higher yields and lower metal losses in the process.

The environmental impact of hydrometallurgical methods is generally lower than pyrometallurgical methods, and energy consumption also tends to be lower [117].

The main disadvantage is the production of large amounts of wastewater and solutions. However, they can be successfully recovered and reused in the process in many cases.

The input for hydrometallurgical processing is usually the active mass or cathode active material of spent portable or industrial LiBs. Currently, LCO-based mass is chemically predominant. NMC- and NCA-based materials are available for recycling to a lesser extent; however, like LCO, they contain cobalt in their structure. There are a number of different leaching systems that are suitable for leaching metals from active mass, e.g., sulfuric acid with added hydrogen peroxide, hydrochloric acid, phosphoric acid with added hydrogen peroxide and others.

The mechanism of leaching metals, especially cobalt and lithium, using sulfuric acid with and without hydrogen peroxide or using hydrochloric acid has been described quite successfully over the years. In both media, quantitative conversions of metals to leachates can be achieved with suitably selected conditions. However, the disadvantage is the low selectivity. Today, research is largely focused on various alternatives for leaching systems, mainly with the aim of reducing costs, increasing selectivity and improving environmental acceptability. The addition of alternative reducing agents to sulfuric acid, such as copper and iron scrap, or the use of leaching agents from the group of organic acids that are generally considered to be more environmentally acceptable, e.g., citric acid, lactic acid and others, are being investigated. Under certain conditions, high yields of the metals of interest can also be achieved in these media. A promising approach is the simultaneous leaching and precipitation of metals using oxalic acid.

The graphite from the anode, which is part of the active mass, is not leached in the above media, but is part of the insoluble residue, from where it can be carefully extracted, e.g., by flotation. However, its re-use in the production of new anode material has a number of limitations, as it is often of insufficient quality. Its structure is very damaged and disordered and the main problem in recycling is to restore the degree of graphitization. Theoretically, spent graphite has three possible uses: regeneration into battery materials, low-value utilization (as reductant, adsorbent) and high-value conversion. At present, its recycling is not carried out in industry, but the possibilities are being investigated at a laboratory scale [74].

The combination of precipitation and solvent extraction in varying order has proven effective for the efficient extraction of individual metals from obtained leachate. When precipitation is used as a method to recover cobalt from the leachate, oxalic acid is one of the effective precipitating agents. The advantages of using oxalic acid are the high precipitation efficiencies and the fact that no additional metal ion is added to the leachate, as is the case, for example, with NaOH precipitation. The disadvantage is the co-precipitation of other present metals, with the exception of lithium. Therefore, it is suitable to precipitate manganese with Na_2_CO_3_ or KMnO_4_ and nickel with dimethylglyoxime prior to cobalt precipitation. Lithium is preferably recovered from solution by precipitation with Na_2_CO_3_ at pH ~12–14.

In the case of solvent extraction, Cyanex 272, Mextral^®^272P and PC-88A are considered as suitable extraction agents for cobalt. The application of ionic liquids and DESs has been a specific recent development, and they have confirmed their potential in the extraction of cobalt with high efficiency. D2EHPA is suitable for the extraction of manganese and lithium; Mextral^®^564OH is suitable for the copper extraction. Under appropriately selected conditions such as suitable pH, phase ratio, extraction time, suitable stripping agent, etc., high efficiencies can be achieved using solvent extraction. The problem with solvent extraction is the requirement to repeat the individual operations in a multi-step process in order to achieve the required efficiency, which makes the extraction process longer and more expensive. The disadvantage of solvent extraction is also the cost of extraction and stripping reagents and the necessity of their regeneration. However, regeneration can achieve cost savings and increase environmental acceptability as well.

Electrochemical processes for spent LiB recycling are a sophisticated method which can include the processes of electrowinning, electrooxidation and electrodeposition of the valuable metal. In addition, electrochemical processes using selective membranes that only transmit one type of ion can also be applied. In this way, the selected metals, e.g., cobalt from lithium, are mutually separated. During electrochemical processes, high metal yields and selectivity can be achieved without the use of aggressive leaching reagents. However, their wider application may be inhibited by relatively high process costs and limited operational capacity.

In the industry, companies combining mechanical-physical pre-treatment and hydrometallurgical processing can be considered as more sophisticated, environmentally friendly and energy efficient. They have usually a lower processing capacity (150–300 tons/year) compared with pyrometallurgical processes. However, the lower processing capacity can partly be considered as an advantage due to the high process flexibility.

Recycling plants that have successfully introduced lithium battery recycling are mainly pyrometallurgical plants that have built on years of experience in processing other types of feedstock and that have adapted existing technologies to the new input materials as spent LiBs, e.g., Umicore. The problem is the safety of the process due to the hazardous nature of spent LiBs and the high energy consumption. Hydrometallurgical plants whose processes are based on “in house” developed methods using sophisticated and highly flexible steps and appropriate mechanical pre-treatment can also be considered successful. A separate group are companies that only use mechanical-physical treatment in their processing, with the benefit of saleable fractions obtained at relatively low input and operating costs. However, in this case it is appropriate if all types of electrochemical cells are the input. For the future, the most promising for portable LiB recycling seems to be hydrometallurgical plants operating today as pilot plants (e.g., OnTO, LithoRec), which provide complex recycling of LiBs with recovery of all recoverable metals and components. These are mainly a combination of leaching with solvent extraction followed by metal electrowinning or precipitation. A major advantage of solvent extraction is the possibility of recovering the organic agent.

## 6. Conclusions

Material recycling of used portable lithium cells has been carried out around the world for several years. However, in the context of their increasing consumption, it is necessary to increase capacities, streamline processes and adapt technologies to other types of spent LiBs, e.g., LiBs from EVs. A comprehensive approach is preferred with the aim of maximizing the recovery of present components. The main challenge in recycling processes is to achieve high selectivity and efficiency of individual processes. The demand for high selectivity results from the heterogeneity of the composition of the spent portable LiBs, which represent a wide mixture of materials such as metal oxides present in the active mass, graphite, copper and aluminum electrodes and a polypropylene separator. Almost all of these components can be recycled in their own cycle after appropriate mechanical-physical pre-treatment. Metals present in the active mass, such as cobalt, magnesium and nickel, pass through the treatment process into shared intermediates with high efficiency, from where they can be selectively extracted. A combination of the following steps occurs in almost every proposed process for complex processing of spent lithium cells: mechanical-physical pre-treatment, thermal pre-treatment, leaching (also multi-stage), precipitation and solvent extraction in several stages and variations and preparation of the final product, e.g., via calcination. EU legislation defines a minimum recycling efficiency of 50% for spent lithium cells and up to 65% in 2025. In the case of pyrometallurgy, where the graphite and separator are combusted and some metals, e.g., lithium and aluminum are lost in the slag, achieving the required recycling efficiency is questionable. From this point of view, hydrometallurgical treatment processes can be considered more appropriate. Hydrometallurgy achieves high metal yields at low losses, and the processes are more flexible and energy efficient. However, the disadvantages include the generation of waste water and waste solutions, as well as the requirement to pre-treat the cells in order to obtain material suitable for leaching.

## Figures and Tables

**Figure 1 materials-16-04264-f001:**
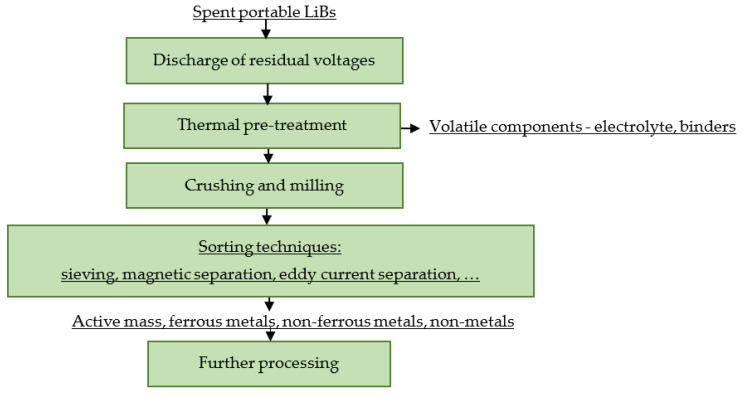
Principle scheme for pre-treatment of spent portable LiBs.

**Figure 2 materials-16-04264-f002:**
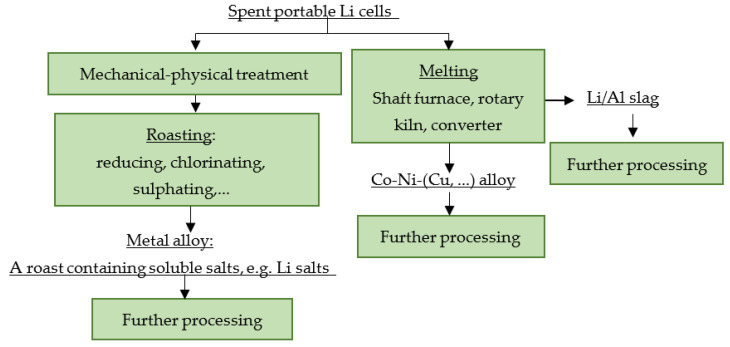
Principle scheme for pyrometallurgical processing of spent portable LiBs.

**Figure 3 materials-16-04264-f003:**
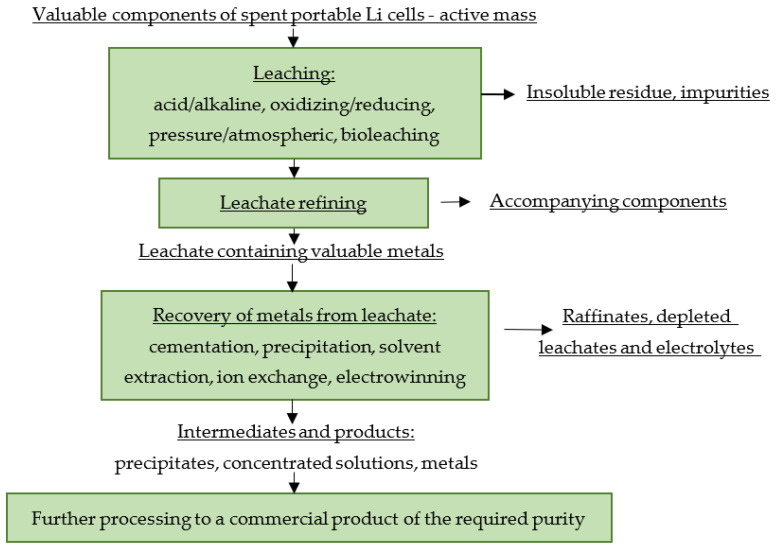
Principle scheme for hydrometallurgical processing of spent portable LiBs.

**Figure 4 materials-16-04264-f004:**
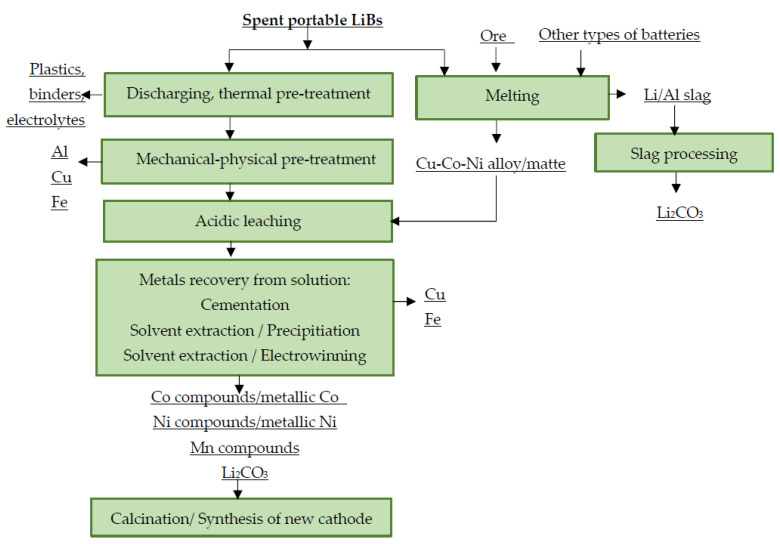
Principle scheme for industrial processing of spent LiBs.

**Figure 5 materials-16-04264-f005:**
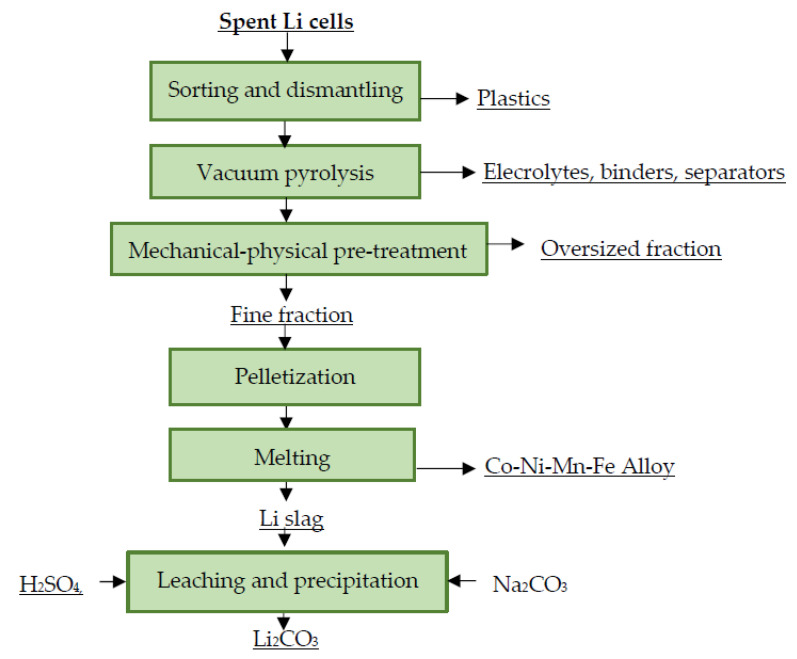
Processing of spent Li cells at Accurec Recycling.

**Figure 6 materials-16-04264-f006:**
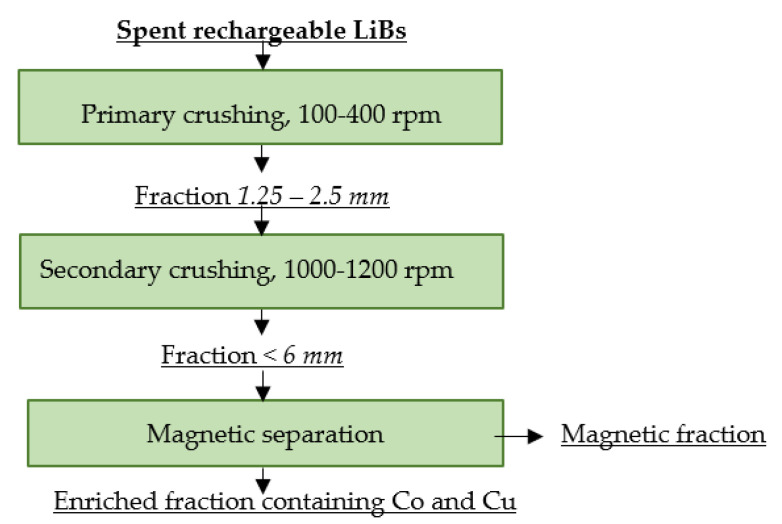
Processing of spent rechargeable LiBs at Akkuser.

**Figure 7 materials-16-04264-f007:**
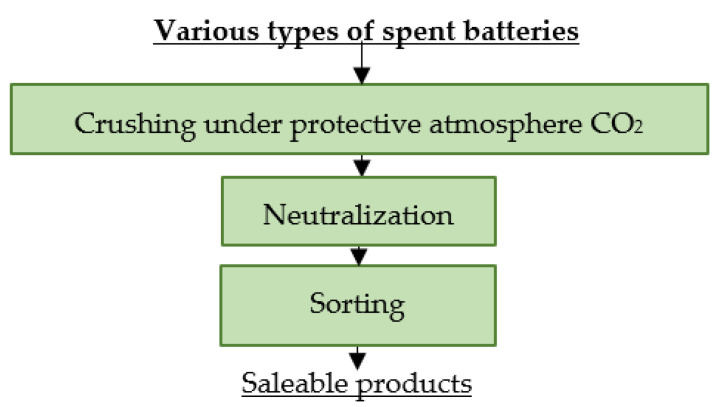
Processing of spent batteries at Batrec AG.

**Figure 8 materials-16-04264-f008:**
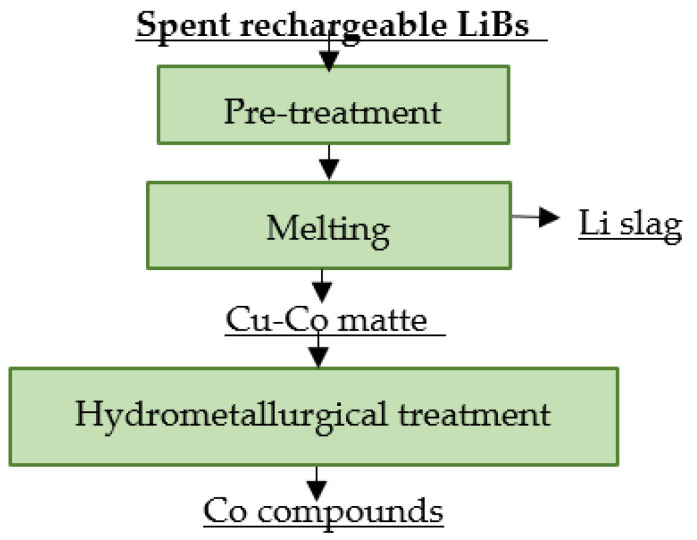
Processing of spent LiBs at Glencore.

**Figure 9 materials-16-04264-f009:**
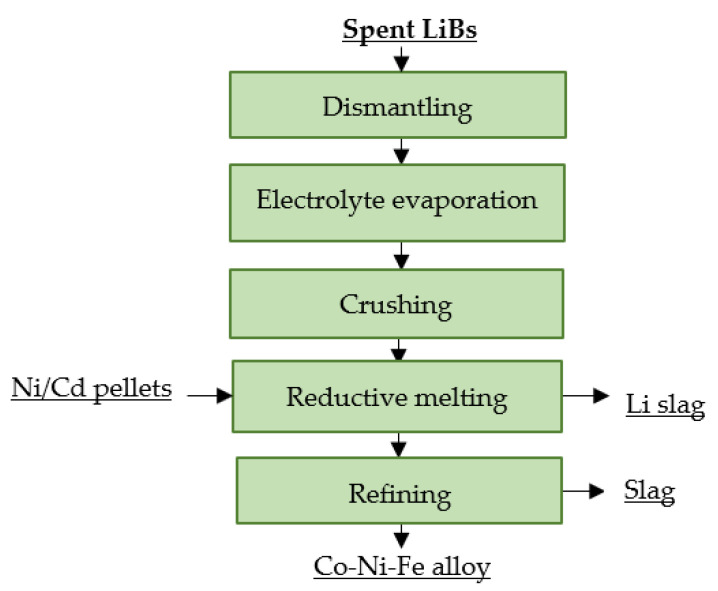
Processing of spent LiBs at Inmetco.

**Figure 10 materials-16-04264-f010:**
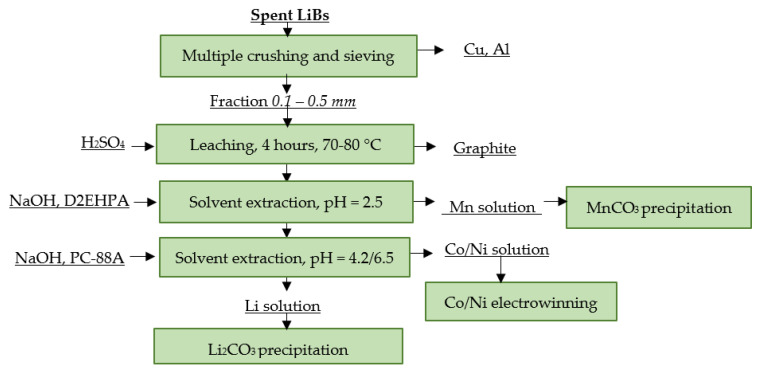
Processing of spent LiBs at JX Nippon Mining and Metals.

**Figure 11 materials-16-04264-f011:**
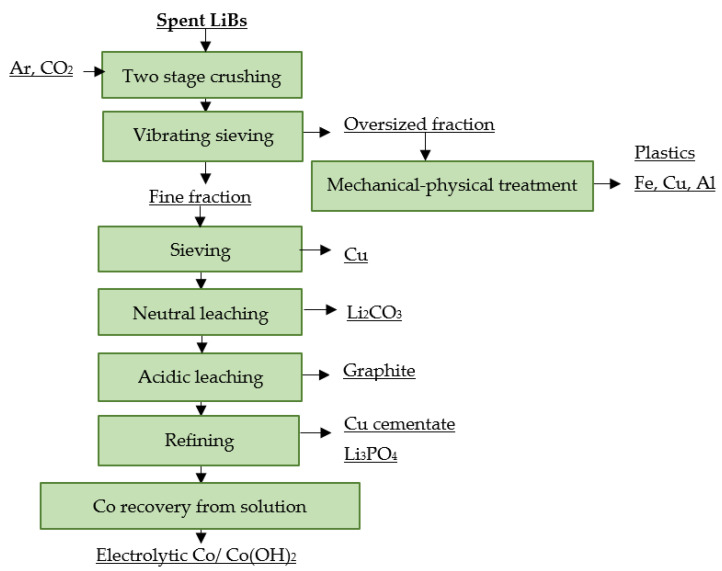
Processing of spent LiBs at Recupyl Valibat.

**Figure 12 materials-16-04264-f012:**
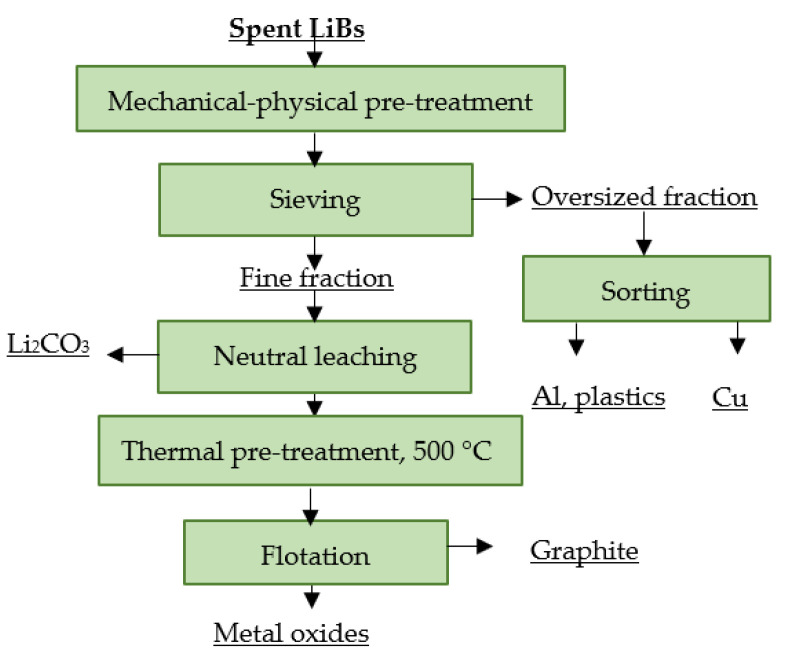
Processing of spent LiBs at Retriev Technologies.

**Figure 13 materials-16-04264-f013:**
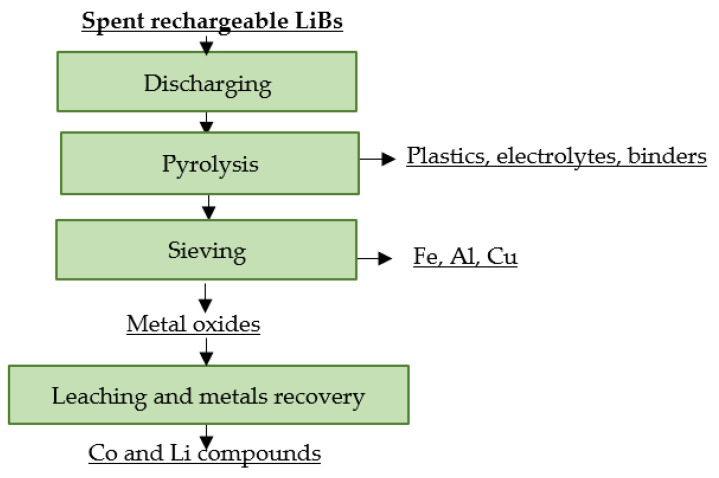
Processing of spent rechargeable LiBs at SNAM.

**Figure 14 materials-16-04264-f014:**
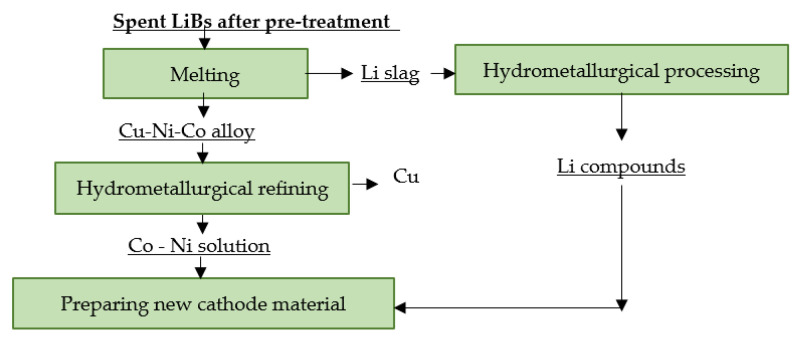
Processing of spent LiBs at SMM.

**Figure 15 materials-16-04264-f015:**
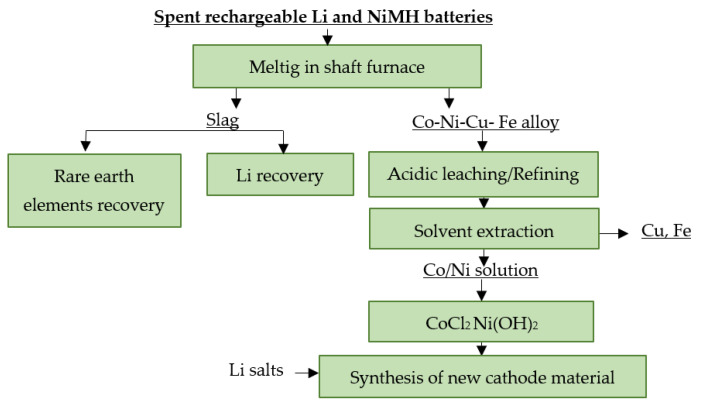
Processing of spent LiBs at Umicore Battery Recycling.

**Figure 16 materials-16-04264-f016:**
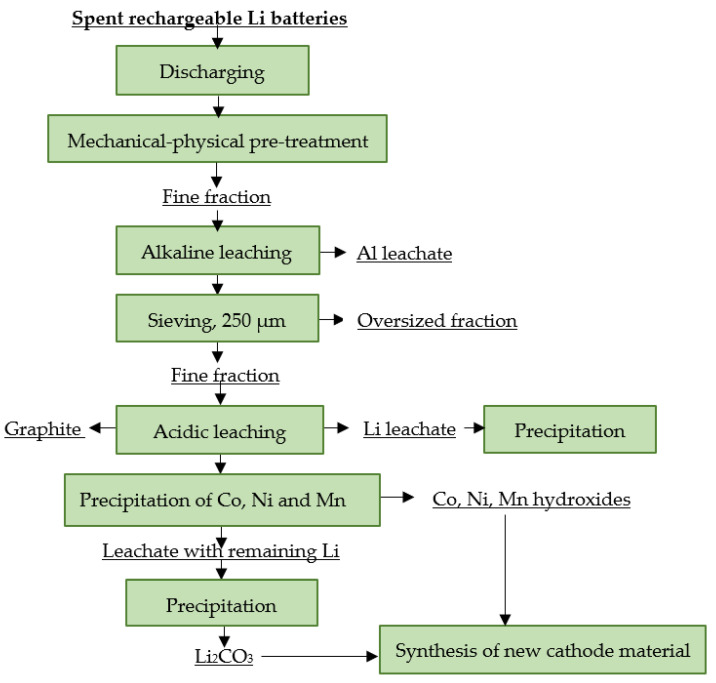
Processing of spent LiBs at BRRP.

**Figure 17 materials-16-04264-f017:**
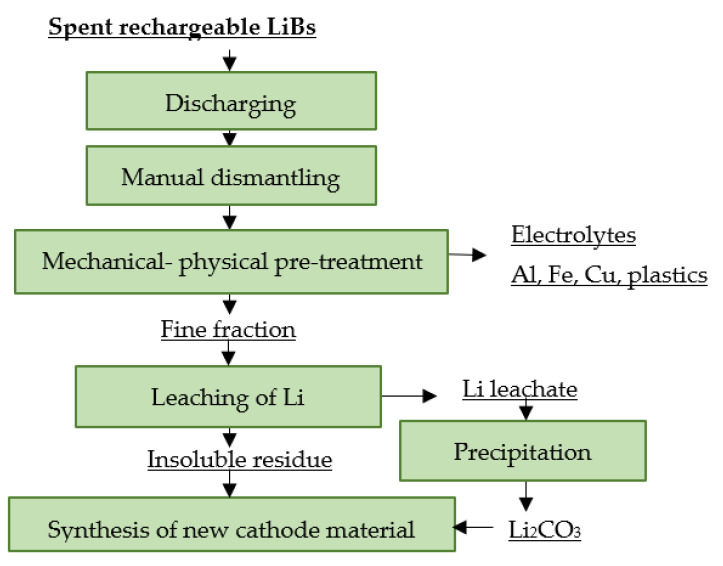
Processing of spent rechargeable LiBs at LithoRec.

**Figure 18 materials-16-04264-f018:**
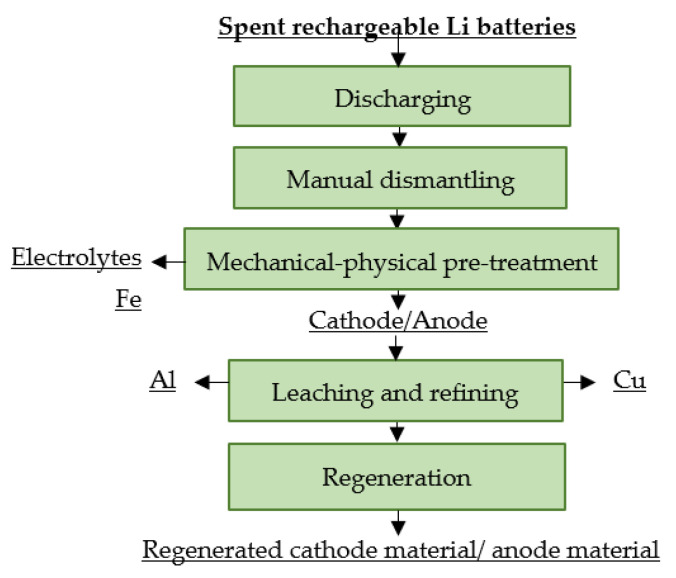
Processing of spent rechargeable LiBs in the OnTo process.

**Table 1 materials-16-04264-t001:** Composition of portable primary LiBs.

Producer	Saft [9]	BiPOWER [10]	Duracell [11]	Ultralife [12]
Type	Li–SOCl_2_ cylindrical	Li–SOCl_2_ cylindrical, prismatic	Li–MnO_2_ cylindrical	Li–MnO_2_ cylindrical, button
Component	Content [%]			
Li	3.5–5	≤5	1–5	3–4
SOCl_2_ or MnO_2_	40–46	≤47	15–45	40–45
AlCl_3,_ other Li salts	1–5	≤5	0–5	1
C	3–4	≤5	0–5	
Ethylene glycol dimethyl ether			5–10	3–4
Propylene Carbonate (PC), Ethylene carbonate (EC)			0–10	4–5
Tetrahydrofuran				5–9

**Table 2 materials-16-04264-t002:** Material composition of portable secondary LiBs [8].

Component of Portable Secondary LiBs	Material	Content [%]
Cover	Steel, aluminum	20–25
Cathode active material	LCO, NMC, NCA, LFP, LMO	25–35
Anode active material	graphite	14–19
Electrolyte	LiPF_6_ dissolved in PC, EC, dimethyl carbonate, diethyl carbonate	10–15
Cathode	Aluminum	5–7
Anode	Copper	5–9
Separator	PP, polyethylene (PE)	1–4
Additives	Carbon black, silicone, etc.	unspecified

**Table 3 materials-16-04264-t003:** Ksp and solubility of compounds of the valuable metals at 25 °C [76,77].

Valuable Metal	Compounds	K_sp_	Solubility in H_2_O [g/L]	
Co	Co_3_(PO_4_)_2_	2.05 × 10^−35^	1.66 × 10^−5^	Insoluble *
	Co(OH)_2_	5.95 × 10^−15^	1.06 × 10^−3^	Insoluble
	CoCO_3_	1.00 × 10^−10^	1.19 × 10^−3^	Insoluble
	Co_2_C_2_O_4_∙2H_2_O (20 °C)	2.70 × 10^−9^	2.12 × 10^−1^	Insoluble
Li	Li_3_PO_4_	2.37 × 10^−11^	9.94 × 10^−1^	Slightly soluble *
	Li_2_CO_3_	8.15 × 10^−4^	4.35	Slightly soluble
Mn	Mn(OH)_2_	2.00 × 10^−13^	3.28 × 10^−3^	Insoluble
	MnCO_3_	2.24 × 10^−11^	5.44 × 10^−4^	Insoluble
Ni	Ni(OH)_2_	5.48 × 10^−16^	4.78 × 10^−4^	Insoluble
	NiCO_3_	1.42 × 10^−7^	4.47 × 10^−2^	Insoluble

* insoluble ≤ 0.1 g/L, slightly soluble = 1–10 g/L [78].

**Table 4 materials-16-04264-t004:** Companies and their technologies for spent LiB recycling [7,109,110,111,112,113,114,115,116,117,118,119,120,121,122,123,124,125,126,127].

Company	Input	Processing	Output	Capacity [Tons/Year]
Accurec Recycling	All types of spent LiBs	Combined	Co-Ni-Mn-Fe alloy; Li_2_CO_3_ from slag	3000
AkkuSer	All types of spent batteries	Mechanical-physical pre-treatment	From spent LiBs— fraction of Co, Cu	4000
Batrec AG	All types of spent batteries	Mechanical-physical pre-treatment	Metals fractions	200
Glencore	Spent rechargeable LiBs	Combined	Co compounds Li slag	7000
INMETCO	Spent rechargeable LiBs	Mechanical-physical pre-treatment Pyrometallurgy	Co-Ni-Fe alloy	6000
JX Nippon Mining and Metals	Waste from cathode active material production	Mechanical-physical pre-treatment Hydrometallurgy	Electrolytic Co, Ni MnCO_3_, Li_2_CO_3_	5000
Recupyl Valibat	Spent rechargeable LiBs	Mechanical-physical pre-treatment Hydrometallurgy	Electrolytic Co Co(OH)_2_ Li_3_PO_4_	110
Retriev Technologies	Spent rechargeable and non-rechargeable LiBs	Combined	LiMeO_2_—new cathode active material Li_2_CO_3_, graphite	4500
SNAM	Spent rechargeable LiBs	Thermal pre-treatment Hydrometallurgy	Saleable products of Co and Li	300
SMM	Spent rechargeable LiBs	Combined	Cathode active material	n.d. **
Umicore Battery Recycling	Spent rechargeable LiBS, Ni-Cd, NiMH batteries	Combined	LiCoO_2_ Ni(OH)_2_	7000
BRRP *	Spent rechargeable LiBs	Mechanical-physical pre-treatment Hydrometallurgy	Cathode active material based on Co, Li, Mn, Ni	n.d. **
LithoRec *	Spent rechargeable LiBs from EV	Mechanical-physical pre-treatment Hydrometallurgy	Cathode active material based on Co, Li, Mn, Ni	2000
OnTo *	Spent rechargeable LiBs from EV	Mechanical-physical pre-treatment Hydrometallurgy	Cathode and anode active material	n.d.

* pilot plant. ** not defined.

## Data Availability

Not applicable.

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
