# Peer review of "Current Trends in Spent Portable Lithium Battery Recycling"

_materials, 2023, doi:10.3390/ma16124264_

Round 1

Reviewer 1 Report

The work submitted for consideration in MATERIALS entitled “Current trends in spent portable lithium batteries recycling” by Zita Takacova , Dusan Orac, Jakub Klimko and Andrea Miskufova tries to give an overlook on the lithium batteries recycling. The first concern is on the “portable” term that should be described, and know clearly the difference, and also announce the advantage of both types.

Minor points:

-Correct in line 341 of Page 9 “7071-73”

-Correct in line 729 “Conclussion”

-In line 737 correct “polypro-pylene

There are more mistakes, for example spaces that should be remove, the journal titles should be abbreviated, after the title of the references sometimes there is a dot sometimes a comma. About the references, they should be updated with the most recent work, basically from 2022 and 2023, including the work of Mentbayeva, Posada-Pérez …

And actually there are more “typos” but I saw them in the first reading I did and I did not mark, and thus a “minor” revision is mandatory, but it is SO easy to read that the readers will enjoy the REVIEW, it will be USER-FRIENDLY.

As stated above there are typos, but not related to the quality of English, actually as stated above, the paper is "user-friendly", fast to read (even though it is long, but of course, since it is a review).

Author Response

Minor points:

-Correct in line 341 of Page 9 “7071-73”

-Correct in line 729 “Conclussion” – line 869

-In line 737 correct “polypro-pylene” – line 878

My comment: everything has been corrected. The references have been edited using MDPI Zotero and abbreviations of journals using Web of knowledge:

https://images.webofknowledge.com/images/help/WOS/R_abrvjt.html

The sources have been added and the whole document has been checked and errors corrected. Thank you

Reviewer 2 Report

The paper is a good synthesis of the processes aimed to the recovery of metals fron spent lithium batteries. In order to improve the readability I suggest to_

1) write a definition of "active mass"

2) dangerous gases or vapours are realesed during thermal treatment?

3) line: 133: organic reagents added to dissolve the binders, are added before or after physical treatment?

4) line 155: what is target metal 29?

5) table 3: Ksp is misleading to indicate the solubility of a soluble compouds. It would be better to report the solubility of each compunds in the table.

Author Response

1) write a definition of "active mass" – lines 83-88

2) dangerous gases or vapours are realesed during thermal treatment? Lines 198-200

3) line: 133: organic reagents added to dissolve the binders, are added before or after physical treatment? – lines 206-208

4) line 155: what is target metal 29? - line 230

5) table 3: Ksp is misleading to indicate the solubility of a soluble compouds. It would be better to report the solubility of each compunds in the table. Thank you for your comment. In the third column, we calculated solubilities using the definition relationship based on the values of Ksp, in the fourth column of the table, we provided solubilities from literary sources. However, we would like to keep the table in the original form due to the fact that the solubilities of individual components in these heterogeneous leaching systems change significantly even with small pH changes. The ionic strength of the solution also has a high impact, increasing the solubility of the components, and of course, solubility is influenced by many other factors. If you decide that the table can remain in its original form, please inform us. We appreciate your understanding.

Reviewer 3 Report

The authors demonstrated an overview of the current state of the field in spent portable lithium battery recycling on both research and industrial scales in this review paper. This work is not a common literature review but still needs a specific view (e.g., Journal of Energy Storage 40 (2021): 102690, Chemosphere 282 (2021): 130944). The above examples also provide a different view for LIB critical review. So, the current work lacks deep thinking. The theme of this content also does not match the scope of MATERIALS.  It is recommended to select a precise viewpoint, then re-submit. 

Here are some questions and suggestions which may improve the quality of this work, as below:

1. Most pre-treatments are reported before. Repeat work is non-senses. Can you provide more details on the specific pre-treatment procedures that are commonly used for spent portable lithium battery recycling? How do these procedures affect the recovery of metals from the batteries?

2. Some illustrations (Schemes or Figures) are required to understand the process well. 

3. What are some of the challenges associated with the recovery of non-metallic materials, such as carbon, from spent portable lithium batteries? How are these challenges being addressed in current research and industrial practices?

4. How do the various pyrometallurgical and hydrometallurgical processes used for spent portable lithium battery recycling compare in terms of their environmental impact and energy consumption?

5. Are there any emerging technologies that show promise for reducing the environmental impact and energy consumption of these processes, improving the efficiency and selectivity of these processes?

6. Provide more information on the existing industrial plants that are focused on spent lithium battery recycling?

7. What are some of the key factors that have contributed to the success of these plants, and what are some of the challenges they have faced?

Author Response

  1. Most pre-treatments are reported before. Repeat work is non-senses. Can you provide more details on the specific pre-treatment procedures that are commonly used for spent portable lithium battery recycling? How do these procedures affect the recovery of metals from the batteries? Lines 709-715, 725-734
  2. Some illustrations (Schemes or Figures) are required to understand the process well. The schemes of each industrial process have been completed according to your requests.
  3. What are some of the challenges associated with the recovery of non-metallic materials, such as carbon, from spent portable lithium batteries? How are these challenges being addressed in current research and industrial practices? Information about Carbon recycling have been added.Lines 382-386, 809-817
  4. How do the various pyrometallurgical and hydrometallurgical processes used for spent portable lithium battery recycling compare in terms of their environmental impact and energy consumption? Lines 785-786
  5. Are there any emerging technologies that show promise for reducing the environmental impact and energy consumption of these processes, improving the efficiency and selectivity of these processes? Yes, there are. It is so-called direct recycling. Information about direct recycling has been added.Lines 770-778
  6. Provide more information on the existing industrial plants that are focused on spent lithium battery recycling? The schemes have been added to each industrial technology. Figure 5-Figure 18
  7. What are some of the key factors that have contributed to the success of these plants, and what are some of the challenges they have faced? Lines 853-868

Reviewer 4 Report

The manuscriptentitled "Current trends in spent portable lithium batteries recycling" by Zita Takacova, Dusan Orac, Jakub Klimko and Andrea Miskufova can be considered with a more or less up-to-date review of the current situation of of the field in spent portable lithium battery recycling on both research and industrial scales.The work provides much useful information that may be useful to researchers working in this area of ​​research.
However, I think the introduction is too concise and the lack of information on the "state of the art".Likewise, as regards the preparation methods, I think that information on the methods in the wet method, which are many and varied, would need to be collected.
On the other hand, I do believe that the discussion is well posed based on the information contained in the manuscript.
This work is ambitious and in my opinion it should include more information in the introduction, synthesis methods and characterization of the materials.  

Author Response

This work is ambitious and in my opinion it should include more information in the introduction, synthesis methods and characterization of the materials.  This information has been added according to your requests in the introduction.

Round 2

Reviewer 3 Report

The revised version looks good. It can be accepted by Materials.

Reviewer 4 Report

The manuscript "Current trends in spent portable lithium batteries recycling"
has been reviewed by the authors Zita Takacova, Dusan Orac, Jakub Klimko,
Andrea Miskufov, the text has been significantly improved. It is also
true that this review about batteries is complex and there is abundant
information and many data about this topic exists.
It seems that this work collects useful information for researchers
in this field
.